# Enteroviruses Manipulate the Unfolded Protein Response through Multifaceted Deregulation of the Ire1-Xbp1 Pathway

**DOI:** 10.3390/v14112486

**Published:** 2022-11-10

**Authors:** Anna Shishova, Ilya Dyugay, Ksenia Fominykh, Victoria Baryshnikova, Alena Dereventsova, Yuriy Turchenko, Anna A. Slavokhotova, Yury Ivin, Sergey E. Dmitriev, Anatoly Gmyl

**Affiliations:** 1Chumakov Federal Scientific Center for Research and Development of Immune-and-Biological Products RAS (FSBSI “Chumakov FSC R&D IBP RAS”), 108819 Moscow, Russia; 2Institute for Translational Medicine and Biotechnology, First Moscow State Medical University (Sechenov University), 117418 Moscow, Russia; 3Belozersky Institute of Physico-Chemical Biology, Lomonosov Moscow State University, 119234 Moscow, Russia

**Keywords:** ER stress, unfolded protein response, Ire1-Xbp1 pathway, picornavirus, poliovirus, coxsackievirus, Ire1 proteolysis

## Abstract

Many viruses are known to trigger endoplasmic reticulum (ER) stress in host cells, which in turn can develop a protective unfolded protein response (UPR). Depending on the conditions, the UPR may lead to either cell survival or programmed cell death. One of three UPR branches involves the upregulation of Xbp1 transcription factor caused by the unconventional cytoplasmic splicing of its mRNA. This process is accomplished by the phosphorylated form of the endoribonuclease/protein kinase Ire1/ERN1. Here, we show that the phosphorylation of Ire1 is up-regulated in HeLa cells early in enterovirus infection but down-regulated at later stages. We also find that Ire1 is cleaved in poliovirus- and coxsackievirus-infected HeLa cells 4–6 h after infection. We further show that the Ire1-mediated Xbp1 mRNA splicing is repressed in infected cells in a time-dependent manner. Thus, our results demonstrate the ability of enteroviruses to actively modulate the Ire1-Xbp1 host defensive pathway by inducing phosphorylation and proteolytic cleavage of the ER stress sensor Ire1, as well as down-regulating its splicing activity. Inactivation of Ire1 could be a novel mode of the UPR manipulation employed by viruses to modify the ER stress response in the infected cells.

## 1. Introduction

Enteroviruses are non-enveloped positive-strand RNA viruses, including a large number of serious human pathogens. The spectrum of diseases caused by these viruses is extremely wide, ranging from acute “common-cold”-like illnesses to poliomyelitis and severe diseases of the central nervous system, heart, skeletal muscles, and liver [1,2,3].

The interaction of a virus with host cytoprotective mechanisms is a key factor underlying its pathogenicity. Upon infection, cells trigger a multifaceted defense response to reduce the negative consequences of viral reproduction and/or activate programmed cell death to avoid further virus spread [4,5,6].

The unfolded protein response (UPR) is one of the most important cytoprotective reactions induced by the accumulation of partially unfolded proteins in the endoplasmic reticulum (ER) that causes ER stress—a situation typical for an infected cell due to the high rate of viral protein synthesis [7,8]. This process may lead, depending on the circumstances, to either cell survival by alleviating the ER stress, or programmed cell death. The interplay of virus infection with this particular cytoprotective mechanism may have a great role in deciding cell fate.

In mammalian cells, three major branches of UPR are activated in response to a misfolded protein influx, mediated by three ER membrane-bound sensors: PERK (RNA-dependent protein kinase-like ER kinase), ATF6 (activating transcription factor 6), and Ire1 (inositol-requiring enzyme 1α, encoded by the *ERN1* gene). The activation of PERK leads to translation attenuation via phosphorylation of the translation initiation factor eIF2α [9]. ATF6 acts as a transcription factor after its proteolytic cleavage in Golgi apparatus upon ER stress; the activated ATF6 triggers the expression of chaperone-encoding genes [10]. Ire1 is a conserved transmembrane protein with an ER–luminal domain that senses misfolded proteins in the ER, most likely by direct ligand-mediated recognition [11,12]. This recognition activates the Ire1 cytoplasmic kinase and endoribonuclease (RNase) domains, which initiate unconventional splicing of an mRNA encoding the transcription factor X-box binding protein 1 (Xbp1) [13]. This switches the Xbp1 mRNA translation to a productive mode and leads to the accumulation of Xbp1s protein isoform (encoded by the spliced Xbp1 mRNA) [12,14]. Xbp1s acts as an activating transcription factor for chaperone-associated genes and those involved in the ER-associated degradation (ERAD) pathway [13]. As an RNase, Ire1 also induces a selective degradation of ER-associated mRNAs: the pathway called RIDD (regulated Ire1-dependent degradation). Along with ERAD, RIDD reduces ER and Golgi overload and maintains secretory protein homeostasis. The process of RIDD-mediated RNA degradation partially relies on the sequence, structure, and translational status of affected mRNAs [15,16,17]. Another important role of Ire1 is the regulation of a switch from the cell survival program to apoptosis. Ire1 triggers the cascade of reactions leading to apoptotic death in cases when the cell is unable to overcome stress conditions [18,19].

During infection, large amounts of viral proteins are synthesized in the infected cell, leading to UPR activation. Different viruses modulate the UPR pathways in order to promote their propagation. Some of them directly activate one, two, or three UPR branches, while others suppress them or even differentially affect the three branches. The Ire1-Xbp1 pathway is the most conservative UPR branch associated with different viral infections [20,21,22].

For example, it was shown that the Xbp1s level is elevated in cells carrying the hepatitis C virus (HCV) subgenomic replicons, while the activity of Xbp1-dependent genes was simultaneously repressed [23]. Another representative of the Flaviviridae family—tick borne encephalitis virus (TBEV)—triggers both the ATF6 and Ire1 branches of UPR, leading to Xbp1s upregulation at both mRNA and protein levels. Cell treatment with 3,5-dibromosalicylaldehyde (an Ire1 inhibitor) significantly represses TBEV replication [24], suggesting the involvement of the Ire1-Xbp1 pathway in the TBEV life cycle. A beneficial effect of Ire1 activation was also shown on Japanese encephalitis virus (JEV) reproduction. The RIDD pathway is activated in JEV-infected cells to a similar extent to that observed under the chemical induction of ER stress. However, despite JEV RNA localization in close proximity to the ER membrane, it is not susceptible to the Ire1-induced cleavage. Moreover, the inhibition of Ire1 RNase activity in infected cells reduces viral titer [25]. The influenza A virus (IAV) activates the Ire1 pathway and upregulates Xbp1 mRNA splicing with little or no concomitant activation of the PERK and ATF6, and inhibition of Ire1 activity leads to decreased viral replication [26]. The adenoviral E3-19K glycoprotein specifically activates the Ire1 nuclease, but not other UPR sensors, initiating mRNA splicing of Xbp1 [27]. There are some other examples of UPR modulation by corona-, flavi-, orthomyxo-, rota-, hepadna-, herpes-, and other RNA and DNA viruses (for review, see [28,29,30,31,32,33,34]).

Three representatives of the Picornaviridae family (*Enterovirus* genera) have been investigated for their ability to modulate the Ire1-Xbp1 pathway—enterovirus 71 (EV-71), coxsackievirus B3 (CVB3), and human rhinovirus 16 (HRV16). EV-71 triggered the phosphorylation of Ire1 at the late stages of infection. An elevated Xbp1 mRNA level was observed in infected cells, yet neither Ire1-mediated Xbp1 mRNA splicing nor the Xbp1s protein were detected [35,36]. CVB3 was also shown to trigger ER stress: upon CVB3 infection, ATF6 and Xbp1 were activated via protein cleavage and mRNA splicing, respectively, but all of these changes occurred at the late stages of infection (12 h post-infection, hpi) [37,38]. In contrast, another enterovirus, HRV16, did not stimulate Xbp1 mRNA splicing and even likely induced the dephosphorylation of Ire1 [39].

Here, we used poliovirus (PV) and CVB3 as model enteroviruses to investigate the infection-induced modulation of the Ire1-Xbp1 pathway in human cells. Our analysis revealed the complicated dynamics of Ire1 autophosphorylation and cleavage during infection, as well as the virus-mediated repression of Xbp1 mRNA splicing in infected cells. These results demonstrate that enteroviruses use multiple mechanisms to extensively manipulate the Ire1-Xbp1 host defensive pathway in the infected cell.

## 2. Materials and Methods

### 2.1. Cells and Viruses

HeLa cells were grown in Dulbecco’s modified Eagle’s medium (DMEM) with 10% fetal bovine serum at 37 °C in a 5% CO_2_ humidified atmosphere. The virus infection was performed as described earlier [40]. One day before infection, ~0.7 × 10^6^ cells were plated onto 35-mm dishes. On the next day, the confluent cells were washed with serum-free medium and either infected with a virus (in 300 μL DMEM, at a multiplicity of infection (MOI) of 40 PFU/cell) or mock-infected (300 μL DMEM), if not indicated otherwise. After 30 min of adsorption with agitation at room temperature (RT), the cells were washed again and incubated at 37 °C in 5% CO_2_ in serum-free DMEM for various times. Where indicated, Q-VD-Oph (Sigma-Aldrich, St. Louis, MO, USA, #SML0063, up to 20 μM), APY29 (Cayman Chemical, Ann Arbor, MI, USA, #22913, up to 180 μM), or MPCMK (Sigma-Aldrich, St. Louis, MO, USA, #M0398, up to 650 μM) were added to the medium at this stage. The viruses used in this work were poliovirus (PV) type I Mahoney and coxsackievirus B3 (CVB3) Nancy strain from the collection of the FSBSI “Chumakov FSC R&D IBP RAS”. All the experiments were performed in a containment environment.

### 2.2. Western Blot Analysis of HeLa Cell Lysates

The HeLa cells were seeded in a 6-well plate and on the next day infected with the specified viruses at an MOI of 40 PFU/cell for 8 h. At indicated time points, cells were lysed in Laemmli buffer. Samples were separated on a gradient (6–12%) SDS-PAGE and transferred onto a nitrocellulose membrane. Then, membranes were blocked in 5% non-fat dry milk in TBS containing 0.05% Tween 20 for 1 h and subsequently incubated with primary antibodies (diluted 1:1000 to 1:2000) at 4 °C overnight. Anti-p-Ire1 monoclonal antibodies (Abcam, Waltham, MA, USA, #ab124945) were used for the detection of the phosphorylated Ire1 form, while anti-Ire1 polyclonal antibodies (Sigma #I6785 or Abcam #ab37073, as indicated) were used for the detection of the total Ire1 protein. Home-made rabbit anti-VP1 antibodies (a dilution of 1:500) and whole rabbit anti-CVB3 serum (1:500) were used for the detection of the PV and CVB3 protein accumulation, respectively, in infected HeLa cells. After washing, the membranes were incubated with an appropriate HRP-conjugated secondary antibody for 1 h at RT and developed using the Clarity ECL substrate (Bio-Rad, Hercules, CA, USA). As a loading control, the blots were probed with an antibody against β-actin (Sigma, USA, #A5316).

### 2.3. RNA Extraction and RT-qPCR

Total RNA was extracted from cells using the TRIzol reagent (Sigma-Aldrich, St. Louis, USA) following the manufacturer’s protocol. The RNA samples were treated with DNase I to avoid genomic DNA contamination. The cDNA was synthesized using M-MLV reverse transcriptase (Invitrogen, Waltham, MA, USA) and random hexamer primers (Evrogen, Moscow, Russia). RT-qPCR was performed with the following primer sets: for the Xbp1 spliced mRNA, forward: 5′-aatgaagtgaggccagtggc-3′, reverse: 5′-tgaagagtcaataccgccagaa-3′, probe: 5′-(FAM)tgctgagtccgcagcaggtgca(RTQ1)-3′; for PV RNA, forward: 5′-ggcagacgagaaatacccat-3′, reverse: 5′-cgaacgtgatcctgagtgtt-3′, probe: 5′-(FAM)ttgattcatgaatttccttcattggca(BHQ1)-3′; for ERdj4 RNA, forward: 5′-agtagacaaaggcatcatttccaa-3′, reverse: 5′-ctgtatgctgattggtagagtcaa-3′. The values were normalized to the level of the *RPL19* transcript obtained with the following primers: forward: 5′- agcggattctcatggaaca-3′, reverse: 5′-ctggtcagccaggagctt-3′, probe: 5′-(FAM)tccacaagctgaaggcagacaagg(RTQ1)-3′. The mock-infected cells treated with 10 mM DTT for 2 h were used as a positive control for UPR activation. The PCR reactions were set up as follows: 5 min at 95 °C, followed by 40 cycles of 20 s at 95 °C and 40 s at 60 °C. The R-412 qPCR kit (Syntol, Moscow, Russia) was used. The data were analyzed with QuantStudio 5 Software (Thermo Fisher Scientific, Waltham, MA, USA). For the PCR experiment followed by agarose gel electrophoresis, the primers 5′-ccttgtagttgagaaccagg-3′ and 5′-ggggcttggtatatatgtgg-3′ were used, producing fragments of either 442 or 416 bp (specific for the unspliced or spliced Xbp1 mRNAs, respectively). In this case, the PCR reactions were set up as follows: 5 min at 95 °C, followed by 40 cycles of 30 s at 95 °C, 30 s at 58 °C, 60 s at 72 °C, and then 7 min at 72 °C.

### 2.4. Data Analysis

Student’s *t*-test was performed using Prism 8 software to compare the two sets of data. A *p*-value of less than 0.01 was considered statistically significant.

## 3. Results

### 3.1. Ire1 Phosphorylation Is Induced in Enterovirus-Infected Cells

To explore whether enteroviruses activate the Ire1-Xbp1 pathway, Western blot analysis was performed to detect the level of phosphorylated Ire1 (p-Ire1) protein in the infected cells. PV type I Mahoney and CVB3 were chosen as representatives of the *Enterovirus* genus. The viruses were added to the HeLa cells at a high MOI (40 PFU/cell, see Materials and Methods) to ensure that every cell was rapidly infected. 

We examined the level of p-Ire1 in the PV- and CVB3-infected cells throughout the whole viral life cycle. The cells were harvested at various time points. Mock-infected cells treated with DTT were used as a positive control. DTT is a well-known ER stress inducer, which disrupts S-S bond formation, leading to the accumulation of unfolded proteins in the ER. The immunoblot analysis with anti-p-Ire1 antibodies (Figure 1, Appendix A) revealed a robust phosphorylation of Ire1 at 4 hpi in both the PV- and CVB3-infected cell, followed by a reduction of the p-Ire1 protein level. This phosphorylation was dependent on the Ire1 kinase activity, as it was attenuated by APY29 [11], a specific ATP-competitive inhibitor of Ire1 (Appendix A).

To compare the kinetics of the Ire1p phosphorylation with a time-course of the virus infection, we analyzed viral protein accumulation in the same samples (Figure 1c). Abundances of both PV and CVB3 VP1 proteins reach their maximum at 4–6 hpi, which coincides with the downregulation of p-Ire1 protein in the infected cells. Thus, we concluded that Ire1 autophosphorylation is induced at the middle stage of enterovirus infection.

### 3.2. Proteolytic Cleavage of Ire1 during the Middle Stage of Enterovirus Infection

We then analyzed the total Ire1 protein levels in the PV- and CVB3-infected cells. The same samples as in Figure 1 were blotted and hybridized with polyclonal anti-Ire1 antibodies from two different sources (Figure 2). 

Surprisingly, with polyclonal antibodies from Sigma (#I6785), in addition to the expected signal of 110 kDa (corresponding to the full-length Ire1 protein), we observed another band of ~60 kDa, which appeared at 4–6 hpi and later in the case of both the PV- and CVB3-infected cells (Figure 2A,B). Considering the fact that this band appeared at the middle stage of infection and increased throughout the poliovirus cycle, it may be a product of Ire1 cleavage by a viral protease or some cellular enzyme activated at this stage.

The same samples were analyzed with another anti-Ire1 antibody (Abcam, #ab37073, raised to the C-terminal end of the protein). Again, we observed the accumulation of an additional band starting from 4 hpi in the PV-infected HeLa cells (Figure 2C). However, in this case the molecular weight of the detected product was slightly different in size, ~70 kDa. 

To reveal whether this cleavage product is specific to enterovirus-infected cells or appears also during other picornavirus infections, we performed the same experiment with HeLa cells infected with encephalomyocarditis virus (EMCV), a member of the *Cardiovirus* genus. As in previous experiments, the cells were infected at an MOI = 40 PFU/cell and harvested every 2 hpi. Immunoblot analysis was performed with the same (Abcam #ab37073) polyclonal antibodies against Ire1 (Figure 2D). In this case, we did not observe any product of proteolytic cleavage. This fact argues for the putative direct or indirect involvement of a specific enteroviral protease in Ire1 cleavage in the infected cells.

It was shown previously by Genentech Inc. that Ire1 may be a target of caspases. In hematopoietic cells, ER stress led to the caspase-mediated cleavage of Ire1 within its cytoplasmic linker region, generating two Ire1 fragments: a ~55 kDa product comprising the ER-lumenal domain and transmembrane segment, and the remaining part of ~50 kDa, as well as some minor products that were visible at ~85 kDa and beyond 49 kDa [41]. To check the hypothesis that Ire1 proteolytic cleavage in the enterovirus-infected cells may be caspase-mediated, we repeated our experiments in the presence of Q-VD-Oph, a potent pan-caspase inhibitor. Protein lysates at different time-points post-infection were analyzed on an immunoblot with polyclonal anti-Ire1 antibodies (Figure 3). We did not observe any difference in the treated and untreated infected cells, as the ~70-kDa product was visible in both cases. In contrast, in a control experiment Q-VD-Oph completely abrogated a staurosporine-induced caspase-9 conversion into p35 activated product (Appendix A). We concluded that the proteolytic cleavage of Ire1 is not mediated by a caspase. Instead, it may be induced by other protease(s) of cellular or viral origin.

### 3.3. Xbp1 mRNA Splicing Is Not Activated in Enterovirus-Infected Cells

To assess whether PV and CVB3 infection triggers unconventional Xbp1 mRNA splicing, we analyzed the level of spliced Xbp1 mRNA in HeLa cells where these viruses induced ER stress and Ire1 phosphorylation. 

Specific primers for real-time qPCR analysis of the spliced Xbp1 mRNA isoform were designed. HeLa cells were infected with PV, the total RNA was isolated at 0, 2, 4, and 6 hpi, followed by RT-qPCR analysis. RNA from the HeLa cells treated with DTT was used as a positive control in these tests. As expected, the DTT treatment increased the level of the Xbp1s mRNA. However, the analysis did not reveal an increase in the spliced Xbp1 mRNA level over the course of PV infection (Figure 4A). Similar results were obtained with CVB3 (Figure 4B). Thus, despite the autophosphorylation of Ire1, its RNase activity needed for unconventional splicing of the Xbp1 mRNA is likely suppressed in the enterovirus-infected cells. In accordance with this, the expression of an Xbp1 downstream target, ERdj4/*DNAJB9*, was not activated in the infected HeLa cells (Figure 4C,D).

### 3.4. Ire1-Mediated Xbp1 mRNA Splicing Is Inhibited in Enterovirus-Infected HeLa Cells in a Time-Dependent Manner

To verify the hypothesis that Xbp1 mRNA splicing is inhibited in enterovirus-infected cells, we analyzed the effects of PV and CVB3 infections on the levels of the Xbp1s mRNA isoform under the conditions of chemically-induced ER stress. The HeLa cells were infected with PV or CVB3 at an MOI = 40 PFU/cell. At 3 hpi, the mock-infected and virus-infected cells were treated with 10 mM DTT to induce ER stress. After 2 h incubation, the total RNA was extracted, and the Xbp1s mRNA levels were measured by RT-qPCR.

As expected, the level of the spliced Xbp1 mRNA greatly increased (>15-fold) in the mock-infected cells after the chemical treatment. However, under the same conditions the Xbp1s mRNA level was only slightly elevated in both the PV- and CVB3-infected cells (Figure 5A). We concluded that even though Ire1 is phosphorylated at the middle and later stages of enterovirus infection, its activity in Xbp1 mRNA splicing is simultaneously inhibited in the infected cells in a virus-dependent manner. This conclusion was confirmed by the analysis of two RT-PCR products corresponding to the unspliced and spliced Xbp1 mRNA isoforms by gel electrophoresis (Appendix A).

Although these results indicated that enterovirus-infected cells are unable to appropriately develop UPR when it is induced at the middle stage of infection (3 hpi), it was unclear whether this inhibitory effect is also present at earlier stages. To clarify this issue, we performed the following experiment. HeLa cells were infected with PV (MOI = 40 PFU/cell), and immediately after infection DTT was added to the growth medium. The DTT treatment did not significantly affect the course of infection, as confirmed by RT-qPCR of the PV genomic RNA (Appendix A).

In contrast to the previous experiment, we found no inhibition of Xbp1 mRNA splicing in the DTT-treated infected cells throughout the entire course of infection, as the relative level of the Xbp1s mRNA isoform was roughly similar to that observed in the DTT-treated mock-infected cells (Figure 5B,C).

We also analyzed the effects of DTT addition at the early stages of PV infection (from 0.5 to 3 hpi) using 2% agarose gel electrophoresis of RT-PCR products corresponding to the spliced and unspliced Xbp1 mRNA isoforms. This analysis revealed no inhibition of DTT-induced Xbp1 splicing if the stress was applied before 1.5 hpi, a partial inhibition at 1.5–2 hpi, and pronounced inhibition when DTT was added after 2 hpi (Figure 5D). We concluded that the ability of enteroviruses to inhibit the Ire1 activity cannot be expressed if ER stress is developed at the beginning of infection and likely requires a viral protein(s).

We then tested whether MPCMK, a viral 2A protease inhibitor [42], abrogates the observed effects of the viral infection on the stress-induced Xbp1 mRNA splicing. We found that the addition of MPCMK indeed restored the elevated level of the Xbp1s mRNA triggered by DTT even in the presence of PV infection (Appendix A). It should be noted however that MPCMK strongly affects the PV infection itself and causes a dramatic reduction of virus yield [42], thus its negative effects cannot be a strong argument, leaving open the question regarding the protease involved in the cleavage of Ire1.

## 4. Discussion

During a viral infection, an enormous influx of newly synthesized proteins often leads to ER stress in infected cells and triggers cytoprotective UPR signaling pathways. The ability to modulate this response may be important for productive infection and viral virulence [20,21,22]. Thus, several viruses have been shown to actively manipulate major UPR branches, including the Ire1-Xbp1 pathway, to promote pathogenesis [23,24,25,26,27,28,29,30,31,32,33,34,35,36,37,38].

In this study we found that enteroviruses (poliovirus type I Mahoney and coxsackievirus B3 Nancy) induce autophosphorylation of protein kinase/endoribonuclease Ire1 in infected HeLa cells at the middle stage of infection, but this does not lead to the accumulation of spliced Xbp1 mRNA, the major product of its RNase activity. Moreover, we showed that the spliced Xbp1 mRNA isoform cannot be efficiently produced in these cells even under conditions of chemically induced ER stress (10 mM DTT). Then, we found that Ire1 is proteolytically cleaved at the middle stage of enterovirus infection.

These results allowed us to conclude that enteroviruses can actively manipulate the Ire1-Xbp1 pathway, simultaneously activating the Ire1 kinase and suppressing its RNase activity. Comparison of the complex kinetics of Ire1 phosphorylation (Figure 1 and Appendix A), Ire1 proteolytic cleavage (Figure 2), Xbp1 mRNA splicing (Figure 4 and Appendix A), and the suppression of chemically-induced Xbp1s production (Figure 5) in the PV- and CVB3-infected cells, suggests that the deregulation of Ire1 activity is likely mediated by both the proteolytic cleavage of the enzyme and other virus-dependent mechanisms (Figure 6). We believe that this modulation is part of a viral strategy to combat cellular antiviral defense systems.

Previous findings showed that the Ire1-Xbp1 pathway is induced in EV-71- and CVB3-infected human cells. Ire1 phosphorylation and an upregulated overall Xbp1 mRNA level was detected a few hours after EV-71 infection of RD cells [35,36]. Intriguingly, neither Ire1-mediated Xbp1 mRNA splicing nor the active Xbp1s protein was detected, and its downstream genes were not activated [35,36]. The authors suggested that the viral 2A^pro^ protease may contribute to the decrease in Xbp1s protein synthesis by the cleavage of translation initiation factor eIF4G [35], although no direct experimental support of this hypothesis was provided. These results correlate well with our findings, although we propose the new model explaining the phenomenon through Ire1 proteolytic cleavage (Figure 6), likely by a viral protease. Another group showed that CVB3 infection induces ER stress and activates all three branches of the UPR, including the Ire1-Xbp1 pathway [37,38]. In particular, Nuan et al. [38] reported that the level of the phosphorylated form of Ire1 increased gradually from ~6 hpi or earlier and reached a peak at ~10 hpi or later. They also detected an upregulation of the Xbp1s mRNA splicing starting from ~8 hpi [38]. The authors concluded that the Ire1-Xbp1 pathway was fully activated during CVB3 infection under the conditions they used. An earlier study of CVB3 infection by Zhang et al. did not focus on Ire1 but showed the activation of Xbp1 mRNA splicing from ~8 hpi [37]. These results partially contradict our findings. This could be due to a difference in infection procedures (as we used a high MOI of 40 PFU/cell and agitation, which provided a very efficient and concerted infection of the whole cell population [40]). Indeed, in our hands the CVB3 life cycle ended with total cell death as early as at ~8 hpi, while in the two mentioned studies the infection continued at 12 hpi. It should also be noted that in our study we used the CVB3 Nancy strain, in contrast to the CVB3 CG strain used by Zhang et al. However, other explanations cannot be excluded, so this issue should be the subject of future research.

Probably the most intriguing finding made in our study is the Ire1 proteolytic cleavage, which occurred at the middle stage of enterovirus infection (Figure 2). The Ire1 proteolysis could be a new mode of UPR modulation employed by viruses to abrogate the fully developed ER stress response in the infected cells. The immune role of Ire1 in the virus-induced UPR is well-established, as well as the fact that viruses may hijack this protective mechanism to facilitate their replication [20,21,22]. Besides the UPR, Ire1 RNase activity can be applied to a direct elimination of membrane-bound viral RNAs via RIDD [43]. Activated Ire1 is also able to control cell death [18,19]. Thus, Ire1 proteolysis could be a way to abort all antiviral activities and facilitate virus reproduction.

Our study raises the question of the source of the Ire1-specific proteolytic activity. Recently, it was shown that in hematopoietic cells, ER stress leads to the caspase-mediated cleavage of Ire1 [41]. However, our experiment with the pan-caspase inhibitor Q-VD-Oph (Figure 3) suggests that in the case of PV infection the Ire1 cleavage is unlikely to be caspase-dependent. In the earlier study by Niwa et al. [44] it was found that UPR induction in COS leads to the proteolytic cleavage of Ire1, releasing a ~60 kDa fragment containing the kinase and nuclease domains that accumulates in the nucleus. A similar relocalization of Ire1 was observed in CHO cells and mouse fibroblasts, but not in cells lacking Presenelin-1, a catalytic subunit of the γ-secretase [44], thus suggesting that this intramembrane protease protein complex could be involved in Ire1 proteolysis. A link between γ-secretase and enterovirus infection is an intriguing issue and could be studied in future.

We also hypothesized that a protease cleaving Ire1 in our experiments may be of viral origin. Indeed, picornaviral proteases are major virulence factors with well-known roles in modulating cellular mRNA translation, intracellular transport, signaling, and innate immunity [45,46,47,48]. Hundreds of host proteins are cleaved by enterovirus proteases to facilitate viral reproduction [49]. Ire1 could be one of the targets, although this hypothesis is yet to be proved.

In summary, we showed that the Ire1-Xbp1 pathway is modulated in enterovirus-infected cells through a multifaceted mechanism, including autophosphorylation and proteolytic cleavage of Ire1 at the middle stage of infection that coincides with the inhibition of Xbp1 mRNA splicing. The UPR modulation employed by enteroviruses, particularly the Ire1 modifications, could be the target for future therapeutic interventions.

## Figures and Tables

**Figure 1 viruses-14-02486-f001:**
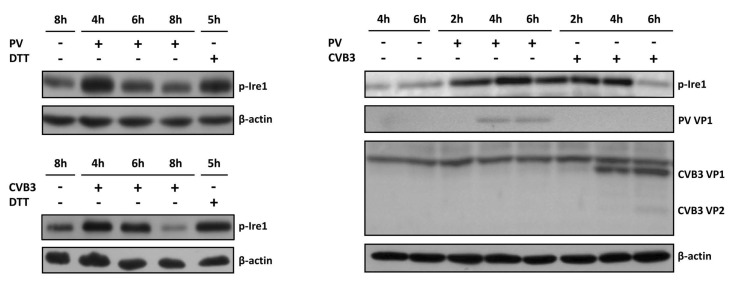
Ire1 phosphorylation in enterovirus-infected cells as revealed by Western blot analysis with anti-p-Ire1 antibodies. (**A**) Phosphorylated Ire1 and total β-actin levels in HeLa cells infected with PV type I Mahoney. (**B**) The same for HeLa cells infected with CVB3. Representative results of at least three independent experiments are shown. (**C**) Accumulation of viral proteins throughout the infection cycle of PV and CVB3 in HeLa cells.

**Figure 2 viruses-14-02486-f002:**
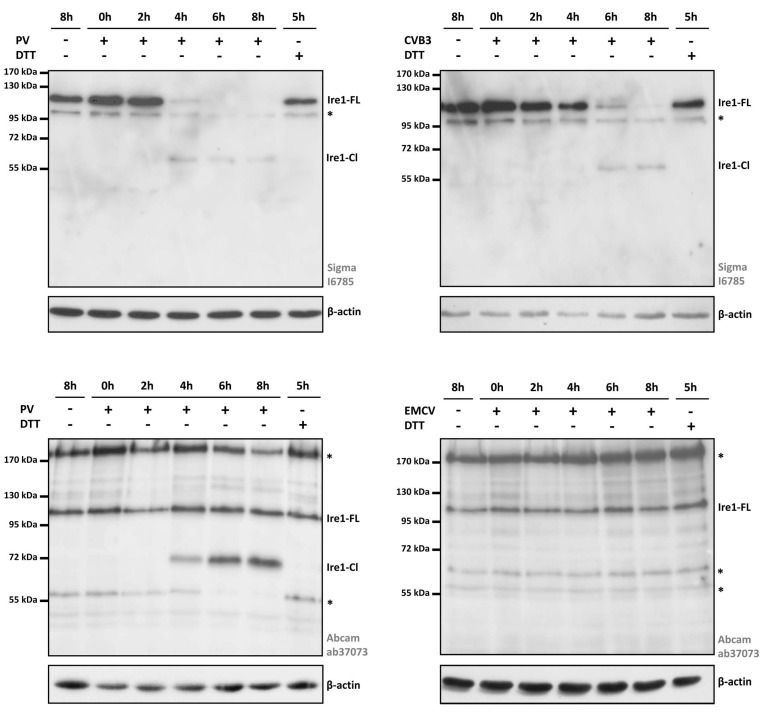
Total Ire1 protein levels in picornavirus-infected HeLa cells analyzed on Western blots using different antibodies. (**A**) Ire1 levels in PV-infected HeLa cells visualized using anti-Ire1 polyclonal antibodies from Sigma (#I6785). (**B**) Ire1 levels in CVB-infected HeLa cells visualized using the same antibodies (#I6785). (**C**) Ire1 levels in PV-infected HeLa cells visualized using an anti-Ire1 polyclonal antibody from Abcam (#ab37073). (**D**) Ire1 levels in EMCV-infected HeLa cells visualized on Western blots using antibodies from Abcam. Representative results of at least three independent experiments are shown. Ire1-FL—the full-length Ire1; Ire1-Cl—products of Ire1 proteolytic cleavage; *—non-specific signals.

**Figure 3 viruses-14-02486-f003:**
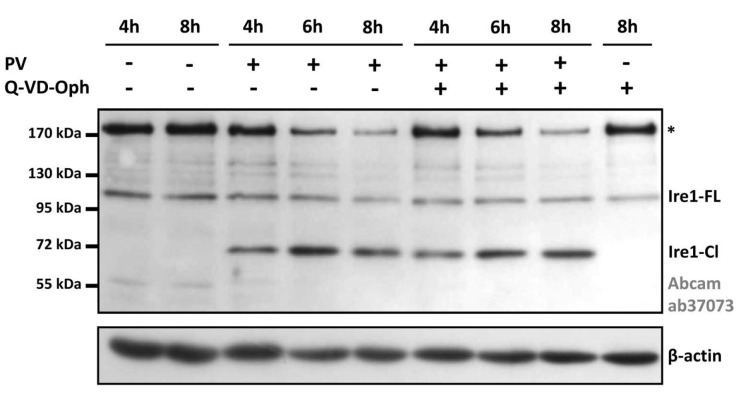
Ire1 proteolytic cleavage in the PV-infected HeLa cells is not affected by the pan-caspase inhibitor Q-VD-Oph. HeLa cells were infected with PV in the absence or presence of 20 μM Q-VD-Oph. After the time indicated, cells were lysed and analyzed by Western blotting with polyclonal anti-Ire1 antibodies (Abcam #ab37073). Asterisks denote the same as in Figure 2.

**Figure 4 viruses-14-02486-f004:**
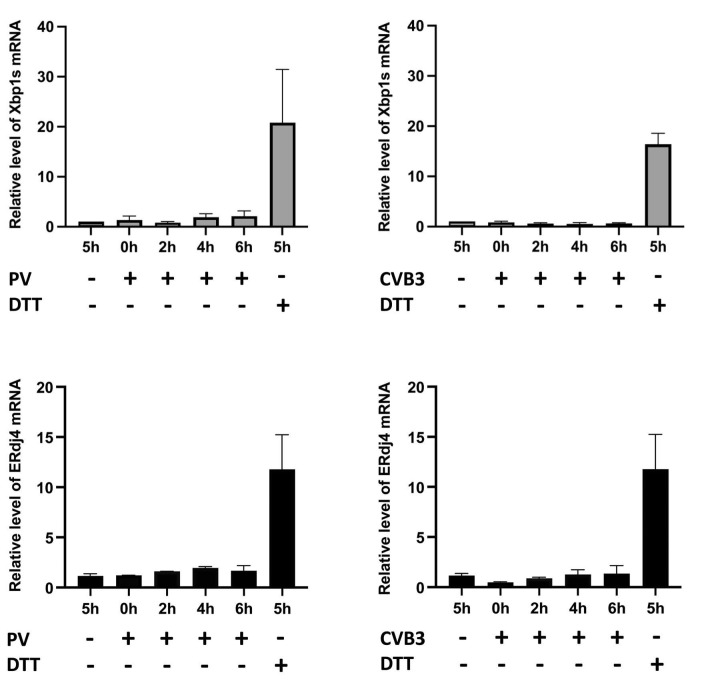
Enterovirus infection does not induce Xbp1 mRNA splicing and activation of the Xbp1 downstream target ERdj4. (**A**) Total RNA from HeLa cells infected with PV, mock-infected, or treated with 10 mM DTT was isolated at the indicated time points. The relative level of the spliced Xbp1 mRNA was measured by RT-qPCR with primers specific to Xbp1s mRNA isoform and to *RPL19* mRNA as a reference. (**B**) The same experiment with CVB3. Each experiment was performed in duplicate and repeated at least three times. (**C**) Total RNA from HeLa cells infected with PV, mock-infected, or treated with 10 mM DTT was isolated at the indicated time points. The relative level of ERdj4 mRNA was measured by RT-qPCR with primers specific to the ERdj4 mRNA and *RPL19* mRNA as a reference. (**D**) The same experiment with CVB3.

**Figure 5 viruses-14-02486-f005:**
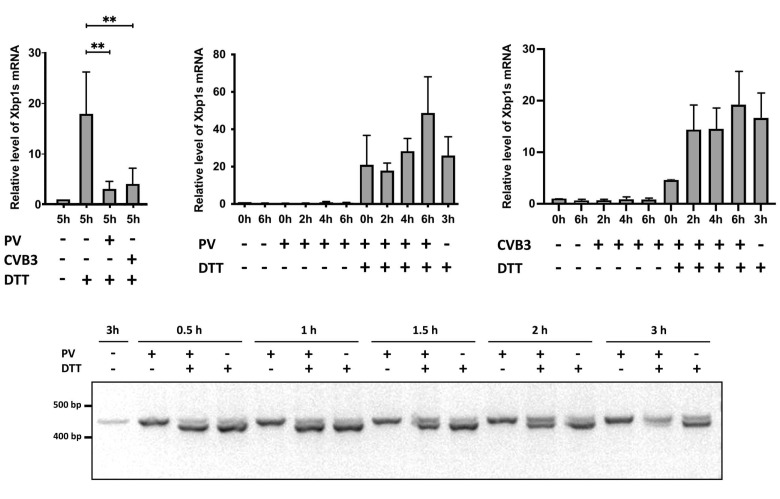
Enteroviruses inhibit Ire1-mediated Xbp1 mRNA splicing at the middle, but not early stage of infection. (**A**) HeLa cells were infected with PV or CVB3, then at 3 hpi 10 mM DTT was added to the medium. 2 h later, total RNA was isolated and the relative levels of the spliced Xbp1 mRNA were measured by RT-qPCR, as described earlier, ** *p* < 0.01. (**B**) Relative Xbp1s mRNA level in PV-infected HeLa cells with and without the addition of 10 mM DTT at 0 hpi, as revealed by RT-qPCR. (**C**) The same experiment as in (**B**), with CVB3. Each experiment was performed in duplicate and repeated at least three times. (**D**) DTT-induced Xbp1 mRNA splicing in PV-infected cells analyzed by agarose gel electrophoresis of RT-PCR products. HeLa cells were infected with PV and treated with 10 mM DTT at the indicated time points. 2 h later, cells were harvested, RNA was extracted, and RT-PCR with an Xbp1-specific primer pair producing fragments of either 442 or 416 bp (corresponding to the unspliced or spliced Xbp1 mRNAs, respectively) was performed. Note that the amounts of the two PCR products should not be compared to each other, as they have different lengths.

**Figure 6 viruses-14-02486-f006:**
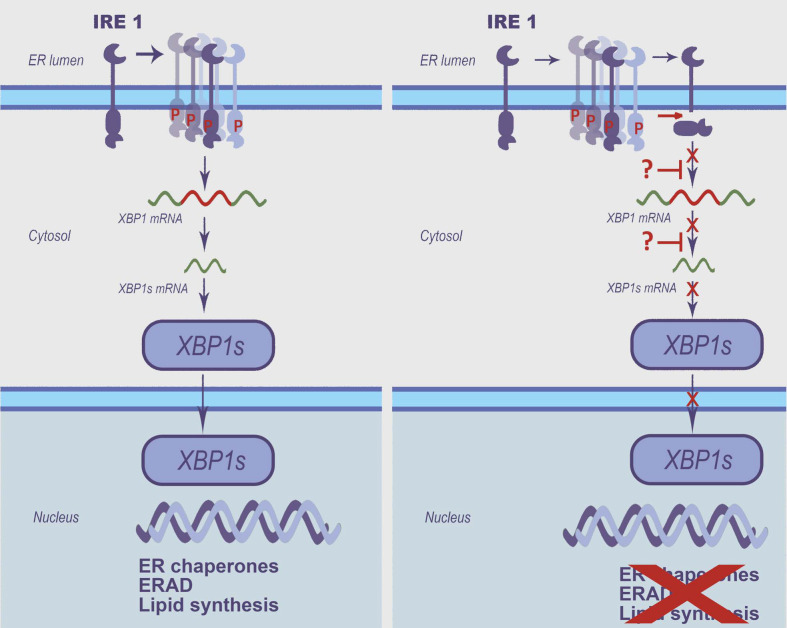
A model explaining the modulation of the Ire1-Xbp1 pathway in enterovirus-infected cells by the induction of Ire1 phosphorylation and inactivation through multiple mechanisms, including proteolytic cleavage. The activated Ire1-Xbp1 pathway in uninfected (**left**) or enterovirus-infected (**right**) cells is shown.

## Data Availability

Not applicable.

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
