# Peer review of "Enteroviruses Manipulate the Unfolded Protein Response through Multifaceted Deregulation of the Ire1-Xbp1 Pathway"

_viruses, 2022, doi:10.3390/v14112486_

Round 1

Reviewer 1 Report

It is well known that the unfolded protein responses (UPR) were hijacked by many viruses including Enteroviruses. In this manuscript, IRE1α-XBP-1 pathway, one branch of three UPRs, was modulated by PV and CVB3 via affecting phosphorylation and cleavage of IRE1α, repressing mRNA splicing of XBP-1 during infection. Although it would be a valuable contribution to the field, this work is not sufficient for publication so far.

1. Who contributes to IRE1α cleavage? It is an interesting question to be answered. Although host caspases are ruled out, viral proteases are not the only candidates. Other host proteases are also potential candidates.

2. XBP-1 mRNA splicing is not activated in CVB3-infected vero cells in this study, but activated in previous study. Some effort needs to be made to explain the difference between the current study and the previous study, such as same MOI of 10.

3. For IRE1α-XBP-1 pathway, more markers should be considered such as XBP-1s protein level and target genes including EDEM.

4. Since IRE1α was phosphorylated by PV and CVB3, why XBP-1 mRNA splicing was undetectable? IRE1α phosphorylation and XBP-1 mRNA splicing should be

detected at the early time of infection.

Reviewer 2 Report

This work by Shishova et al. investigates one unfolded protein response pathway, the IRE1-XPB1 branch, during enterovirus infection. The manuscript describes enterovirus infection dynamically affects the IRE1 pathway by increasing IRE1 phosphorylation in the early stage of infection but reduced during the late stage. The cleavage of IRE1 during viral infection explains these dynamic changes and also provides a way to counteract the ER stress induced by viral infection. 

The finding of IRE1 cleavage during viral infection is novel and explains that enterovirus infection activates IRE1, but its downstream XBP1s were not produced. However, several essential controls were missing as described following:

  1. In Fig. 1 and 3, the expression of viral protein(s), at least one viral protein, should be included. The experiment in Fig. 1-3 is a time course of viral infection. The time of viral protein expression is important information to interpret the results.
  2. The positive control to show the pan-caspase inhibitor did work well in Fig. 3 is missing. If PV infection induces cell apoptosis, it is better to detect cleavage of PARP or caspase three by using the same cell lysate in Fig. 3, and cleaved PARP or caspase three is inhibited by pan-caspase inhibitor. 
  3. Fig. 5 seems not relevant to Fig. 1-3. Since the authors claim that the inhibition of Xbp1s mRNA probably required a viral protein, the expression of viral protein(s) during viral infection under DTT treatment in two different conditions should be included in Fig. 5. In addition, it is better to provide the expression level of phosphorylation of IRE1 or total IRE1/cleaved IRE1 which will connect Fig. 5 result to Fig. 1-3. 

Author Response

Dear Editor,

Thank you for the positive evaluation of our manuscript «Enteroviruses manipulate the unfolded protein response by proteolytic cleavage of Ire1» and for providing us the opportunity to resubmit a revised version. We are very grateful to both Referees for careful reviewing, critical remarks and valuable suggestions, which helped us to improve the manuscript.

In accordance with their comments, we perform a number of additional experiments to support our conclusions, and made corresponding changes in text and figures of the manuscript (shadowed in yellow in the main text as well as here). We believe that the revised version now meets all requirements by Viruses, so we hope that you may find it worthy of publication in the Journal.

Please find below the detailed response to all the comments made by the Referees.

Reviewer #2

  1.  In Fig. 1 and 3, the expression of viral protein(s), at least one viral protein, should be included. The experiment in Fig. 1-3 is a time course of viral infection. The time of viral protein expression is important information to interpret the results.

Thank you very much for this valuable suggestion. We performed the time-course experiment to analyze the production of PV VP1 protein and CVB3 VP1 and VP2 proteins in infected cells throughout the viral reproduction cycles. The results are shown in new Figure 1C.

  1. The positive control to show the pan-caspase inhibitor did work well in Fig. 3 is missing. If PV infection induces cell apoptosis, it is better to detect cleavage of PARP or caspase three by using the same cell lysate in Fig. 3, and cleaved PARP or caspase three is inhibited by pan-caspase inhibitor. 

Indeed, the experiment with the pan-caspase inhibitor lacked this very important control necessary to be sure that it really worked. During the revision, we have solved this problem (see below). However, the aim of this experiment was not to analyze a virus-induced apoptosis but to test the hypothesis that a caspase could be a source of the Ire1 cleavage activity (as previously reported by Shemorry et al., 2019). Thus, to check the ability of the Q-VD-Oph preparation to abort cell death program and inhibit caspase cleavage, we performed a new experiment with staurosporine, a potent apoptosis inducer. As we show in new Supplementary Figure S3, 20 μM Q-VD-Oph (the concentration we used in our original experiment) inhibits production of the active (cleaved) p35 caspase form completely.

  1. 5 seems not relevant to Fig. 1-3. Since the authors claim that the inhibition of Xbp1s mRNA probably required a viral protein, the expression of viral protein(s) during viral infection under DTT treatment in two different conditions should be included in Fig. 5. In addition, it is better to provide the expression level of phosphorylation of IRE1 or total IRE1/cleaved IRE1 which will connect Fig. 5 result to Fig. 1-3. 

The aim of these experiments was to show that the ability of enteroviruses to inhibit the Ire1 activity appears at the middle stage but cannot be expressed if ER stress is developed at the beginning of infection. We agree with the Referee that this conclusion requires confirmation that DTT does not substantially affect the virus life cycle. To confirm this, in the original experiment we had a supplementary figure (which is now Supplementary Figure S6) showing no critical difference in viral RNA accumulation in the presence vs. absence of DTT.

However, after the Referee’s responses were received, we decided that it is important to explore at which time point the inhibition of DTT-induced Xbp1 splicing actually begins. Thus, we analyzed the effects of DTT addition at the early stages of PV infection (from 0.5 to 3 hpi) using gel electrophoresis of RT-PCR products corresponding to the spliced and unspliced Xbp1 mRNA isoforms. This analysis revealed no inhibition of DTT-induced Xbp1 splicing if the stress was applied before 1.5 hpi, a partial inhibition at 1.5-2 hpi, and pronounced inhibition when DTT was added after 2 hpi (new Figure 5D). Although we did not accompany these analysis with an analysis of viral protein accumulation, we would like to note that the experiment in Figure 5D addressed the Referee’s point (3) even better, as in this case we allowed the viral infection to develop in the absence of DTT (i.e. with the same kinetics as in Figures 1-3), then added DTT at different time points and observed a clear difference (see Figure 5D, compare results at 1 h and 3 h).

In addition, as we noted in the response to Referee 1 (see above), we made our conclusions regarding a source of the Ire1 cleavage activity (viral vs. cellular) much more accurate.

Again, we are very grateful to the Reviewer 2 for valuable suggestions and positive evaluations of our manuscript, and believe that the improvements we have made are sufficient to make the positive decision.

Anna Shishova.

Round 2

Reviewer 1 Report

Thanks for the authors' efforts. Based on the current results, PV induces IRE1α phosphorylation before 2hpi (Fig. 1C) and IRE1α cleavage at 4hpi (Fig. 2C), but not XBP-1 RNA splicing from 1hpi to 7hpi (Fig. S5), suggesting that some factor(s) targets IRE1α-mediated XBP-1 RNA splicing at so early stage of infection although IRE1α phosphorylation occurs. It is a very interesting subject, however, the reviewer cannot recommend this manuscript for publication in VIRUSES due to unclear demonstration.

Reviewer 2 Report

The revised version has been much improved. 
